# Analysis of the Dynamics and Characteristics of Rice Stem Tillers via Water Level Management

**Xiujun Hu [1,\*], Yueyang Yu [2], Yuedong Xia [3], Feng Xie [1] and Menghua Xiao [4]**

[1] Zhejiang Institute of Water Resources and Ocean Engineering, Zhejiang University of Water Resources and Electric Power, Hangzhou 310018, China

[2] Zhejiang Qiantang River Basin Centre, Hangzhou 310016, China

[3] Pinghu Water Resources Bureau, Jiaxing 314215, China

[4] Zhejiang Institute of Hydraulics and Estuary (Zhejiang Institute of Marine Planning and Design), Hangzhou 310020, China

[\*] Correspondence: huxj@zjweu.edu.cn; Tel.: +86-18268108215

**Abstract:** Based on theoretical analysis and numerical calculations, this study systematically investigated the changes in rice tillering dynamics and the simulation of stem tiller growth during the tillering stage using the farm water level as a regulation index for rice irrigation and drainage. Based on pit testing, the results of this study show that both flooding and drought in the tillering stage suppress the tiller output of rice and have a certain compensating effect following rehydration. Heavy drought during the tillering period reduced the effective tiller rate, while flooding and light drought had little effect on the effective tiller rate. Flooding and maintaining a high infiltration rate also increased the effective tiller rate. The primary kinetic model of tiller elongation (DMOR) was a good fit for the tiller elongation process (coefficients of determination of 0.99 or higher). In addition, the growth and extinction rates of the stem tiller extinction curves were fitted. The maximum growth rate of the stem tiller growth segment was ranked as CK > L1 > H1 > L2 > H2, and the maximum extinction rate of the stem tiller extinction segment was ranked as CK > H2 > H1 > L2 > L1, indicating that both flooding and drought during the tillering stage could reduce the growth and extinction rates of the stem tiller. This shows that both flooding and drought can reduce the growth and extinction rates of tillers.

**Keywords:** flooded; drought; stem tiller dynamics; kinetic model DMOR

## 1. Introduction

Rice is native to swampy areas in the tropics and subtropics. In the long-term phylogenetic process, it has formed a unique aerenchyma and root structure, which gives rice a dual adaptability to water and drought. However, its waterlogging and drought tolerance must also be within a certain range of the water level. If the water level is too high or too low, its growth and development will be blocked, and its yield will be reduced, which directly leads to plant death in serious cases.

Individual rice shape is mainly determined by the genetic basis, while population characteristics are mainly controlled by habitat factors [1–3]. The development of the quality of the rice root, stem, and leaf population determines the quality of the rice population [4–6]. Tillers are the branches of the rice plant, and their occurrence is an important indicator of individual fitness. At the same time, the tiller power and tiller success rate are important factors affecting rice yield [7,8]. The supply of nutrients to tillers from the main stem of rice is limited by nodulation. Under suitable growth conditions, the main stem has more nutrients for tiller growth before nodulation [9,10]. At the later stage of nodulation, the main stem requires several nutrients due to the rapid growth of the stem, spike, and leaves; therefore, the supply of material to the tillers rapidly decreases. At this point, a tiller with less than three leaves cannot supply nutrients independently and stops growing due to

a lack of nutrition; this is known as an ineffective tiller [11,12]. Therefore, the dynamics of rice stem tillers is a combination of the production of new tillers and the extinction of ineffective tillers. Water level management is the effective control of rice growth and development via the regulation of water, fertilizer, air, and heat in the field [13,14].

Research on rice stem tillers and growth indicators takes three main directions, one of which is to study the effect of different fertilization measures on rice stem tiller dynamics. De Datta showed that N fertilizer application in the early stage of rice fertility mainly affected the number of tillers, and N fertilizer application in the later stage mainly affected the number of grains per spike and the fruit set rate [15]. Chen Shuqiang et al. showed that, at a low N level, increasing the proportion of spike grain fertilizer application could significantly reduce the population stem tiller number and lead to an insufficient spike yield. At a medium-to-high N level, appropriately increasing the proportion of spike grain fertilizer application could improve the effective tiller rate and spike yield [16]. Secondly, the effect of agronomic measures on rice stem tiller dynamics was studied. Li J et al. concluded that the total number of tillers occurring in direct-seeded rice was large compared to those occurring in hand-planted rice, but the spike rate was low [17]. Chao Zhang et al. showed that improving the photosynthetic performance of leaves via cultivation measures, such as improving the direct seeding method, can be beneficial for rice yield [18]. Thirdly, in studying the effect of irrigation methods on rice stem tiller dynamics, Zhou Mingyao et al. showed that the effect of "fertilization and water regulation" in rice under soil water stress was influenced by the degree of soil drought and the amount of nitrogen applied [19,20]. Yu Shuang'en et al. showed that the interactive effect of irrigation and N application had a significant effect on plant height and the number of stem tillers in rice and that additional N fertilization could, to a certain extent, alleviate the decline in plant height and the number of stem tillers caused by water stress [21]. Rice grows in both hot and rainy seasons. Due to heavy rainfall, fields can be subjected to severe flooding, and their crops cannot be harvested in time, or the surrounding water level jacking causes a waterlogging disaster. On the other hand, when the groundwater level drops to a certain depth, the soil water in the root layer cannot meet the requirements of the crop root system to absorb water. At this time, rice growth suffers from drought due to water stress. The problems of waterlogging and drought in rice are caused by changes in farmland water levels [22,23].

Farmland water level refers to the depth of the water layer in paddy fields after rainfall or irrigation and the buried depth of the groundwater in a paddy field when there is no water layer. When the groundwater level is at the surface, the water level of the field is 0. When there is a water layer, the water level of the paddy field is positive, and when there is no water layer, the water level of the paddy field is negative. The strategy of maintaining an appropriate water level on the field surface or the appropriate buried depth of field groundwater through irrigation and drainage measures is called farmland water level regulation. It is easier to observe and master the regulation indicators of irrigation and drainage in practice, which has important theoretical value and practical significance [24,25].

In this paper, the farmland water level was used as the regulation index of rice irrigation and drainage. A field experiment was carried out to study the dynamic change characteristics of the stems and tillers of rice using farmland water level management (waterlogging and drought), which is important for investigating changes in rice population characteristics under different water levels. This study was conducted to find out how water level management affects the tillering process of rice during the critical period of tillering, with the aim of guiding the developments of reasonable water level control standards for rice irrigation and drainage.

## 2. Materials and Methods

### 2.1. Basic Information about the Pilot Area

The experiment was conducted from June to October 2012 at Hohai University. The test area has a humid subtropical climate, with an average annual rainfall of 1021.3 mm, an average annual evaporation of 900 mm, an average annual temperature of 15.7 °C, a maximum monthly average temperature of 28.1 °C, an average annual sunshine duration of 2212.8 h, and an annual frost-free period of 237 d. Rainfall during the flood season (May to September) accounts for more than 60% of the average annual rainfall. The test area is equipped with 32 stationary evapotranspiration measuring pits (28 with a bottom and 4 without a bottom), each of which measures 2.5 m × 2.0 m × 2.0 m (length × width × depth), the above-ground part is equipped with solenoid valve equipment for automatic irrigation, and the underground part is connected with a corresponding large water column, which can realize the automatic control of the underground water level of the measuring pits through internal probes. The relevant physico-chemical parameters of the 0–30 mm soil in the measuring pit are as follows: bulk weight 1.46 g/cm$^3$, pH value 6.97, total nitrogen content 0.9048 g/kg, fast-acting nitrogen content 27.65 mg/kg, total phosphorus content 0.32 g/kg, fast-acting phosphorus content 12.5 mg/kg, soil organic matter content 2.40%, and field water holding rate 25.28%. The geographical location of the test area is shown in Figure 1.

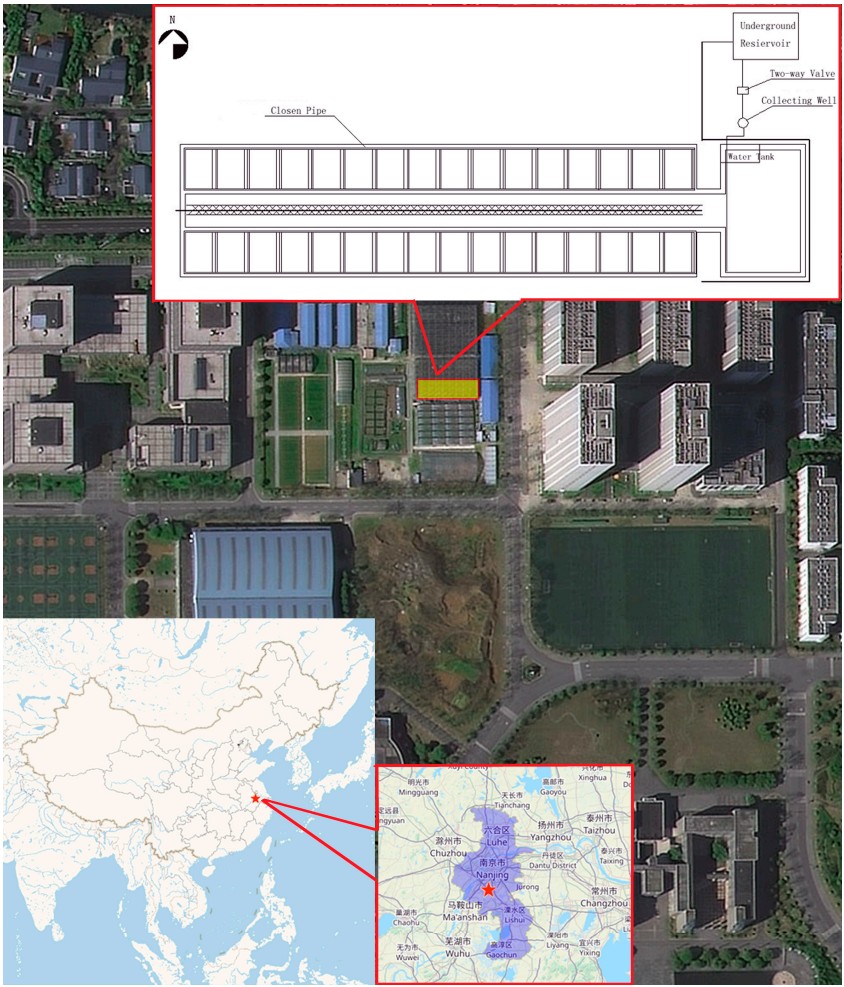

**Figure 1.** Map of the study area.

### 2.2. Experimental Design

In this paper, the farmland water level is taken as the technical index of paddy field irrigation and drainage regulation, coupled with water-saving irrigation and controlled drainage technology. This index involves the dual adaptation characteristics of rice to water and drought to exert drought-tolerance characteristics in the irrigation process, to determine the drought-tolerance limit, to exert flood resistance characteristics in the process of heavy rainfall, and to seek the flood tolerance limit.

Rice that is deeply flooded and maintains a certain intensity of seepage is considered to be completely flooded. The flooded trials were designed with two seepage intensities: 2 mm/d and 4 mm/d. Rice plants are shorter in the tillering stage. The flooding water level was designed to be 120 mm during the tillering stage and was maintained for 10 d. The water level was not replenished to simulate the dynamic change in the water level in the paddy field after a short period of heavy rainfall. The drought test used the buried depth of groundwater as the regulated water level, with drought water levels of −300 mm and −500 mm during the tillering stage. The drought test was a natural descent from the anhydrous layer to a set value to simulate dynamic changes in groundwater levels. The water level management scheme is shown in Table 1.

**Table 1.** Rice water level management trial program.

| Processing Number | | Regulating Water Levels at All Fertility Stages | | | |
|---|---|---|---|---|---|
| | | Tiller Stage | Growth and Gestation Period | Spike Flowering Period | Mammary Period |
| Flooded (10 d) | L1 | 120 mm (2 mm/d) | −300 mm to 30 mm | −300 mm to 30 mm | −300 mm to 30 mm |
| | L2 | 120 mm (4 mm/d) | −300 mm to 30 mm | −300 mm to 30 mm | −300 mm to 30 mm |
| Drought-affected | H1 | −300 mm | −300 mm to 30 mm | −300 mm to 30 mm | −300 mm to 30 mm |
| | H2 | −500 mm | −300 mm to 30 mm | −300 mm to 30 mm | −300 mm to 30 mm |
| Compared to | CK | −200 mm to 20 mm | −300 mm to 30 mm | −300 mm to 30 mm | −300 mm to 30 mm |

### 2.3. Observation Indicators and Analysis Methods

Stem tiller indicator observation: five holes of representative rice stems were selected in each pit for tagging and labeling, and the dynamics of the stems were observed at fixed points: once every 5 days during the tillering period and once every reproductive period thereafter.

Stem tiller elongation model: this paper uses a basic dynamic model to describe the whole process of the growth and decline of the rice population (DMOR). The model can better describe the whole process of stem tiller elongation, and the model parameters have a precise biological meaning. The model expressions are:

$$N = N_A - N_B + C = \frac{A}{1 + c_1 \cdot e^{-b_1 \cdot t}} - \frac{B}{1 + c_2 \cdot e^{-b_2 \cdot t}} + C \tag{1}$$

where $N$ is the number of rice population tillers at time $t$, plants/hm$^2$; $A$ is the maximum possible number of rice population tillers; $B$ is the maximum number of rice population tillers that must die; $C$ is the base parameter, i.e., the basic amount of tillers that does not change with time; $b_1$, $c_1$, $b_2$, and $c_2$ are control parameters; and $c_1$ and $c_2$ are used to characterize the growth rate and extinction rate that tend to be near maximum levels, respectively.

## 3. Analysis of the Results

### 3.1. Changes in Rice Stem Tiller Dynamics in the Tillering Stage via Water Level Management

The dynamics of a rice stem tiller and its daily growth via water level management are shown in Figure 2. It can be seen that the number of tillers in flooded and drought-affected rice was significantly lower than that in the control treatment, but the peak tiller number was reached at essentially the same time in all treatments. In terms of the daily

increase in the tiller number, all rice treatments maintained a high value of growth during the water control period, and the daily growth of stem tillers was higher in the control treatment than in the flooded and drought-affected rice. This indicates that both flooded and drought-affected rice during the tillering period inhibit the rate of tiller emergence, which are two important ways to control tillering in rice. When normal water levels were restored, rice tillering was nearing saturation and daily growth was decreasing in all treatments. Compared to the control, the daily growth of flooded and drought-affected rice decreased at a slower rate and to a lesser extent. The daily growth of drought-affected rice exceeded that of the control after 2 d of rehydration, and the daily growth of the flooded rice exceeded that of the control after 3–4 d of rehydration. This indicates that both flooded and drought-affected rice showed some compensatory effect after rehydration, increasing the rate of rice tillering, with the compensatory effect being more pronounced in drought-affected rice.

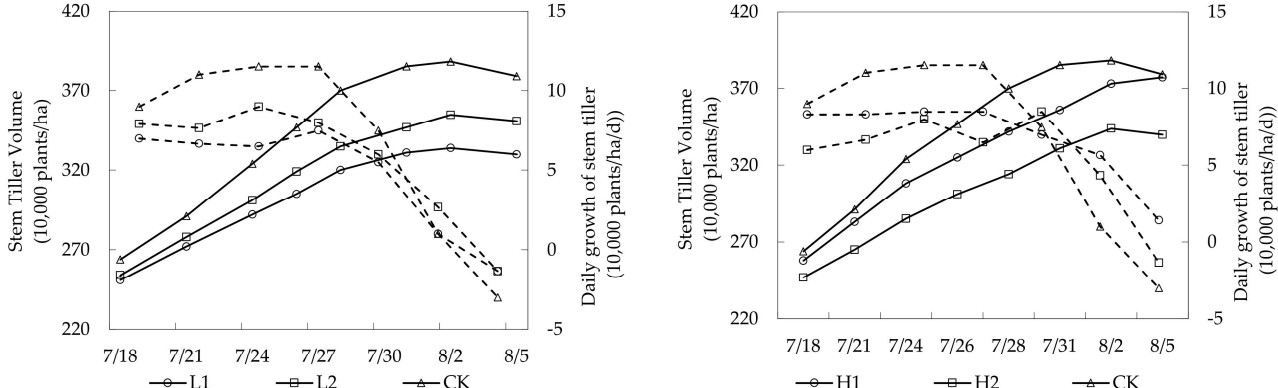

**Figure 2.** Dynamics of rice stem tiller volume and daily growth during tiller stage water level management.

*3.2. Analysis of Stem Tiller Characteristics of Rice during the Tillering Stage via Water Level Management*

The spike rate of rice tillers is an important indicator of group quality, and there is a close relationship between tiller characteristics and yield and its constituent factors. A stable number of spikes and a high rate of spike formation are conducive to improving canopy structure and group quality, improving light conditions for the group in the middle and late stages, increasing the photosynthetic efficiency of the group after tasseling, and increasing dry matter accumulation. The characteristics of rice stem tillers that are affected by water level during the tillering stage are shown in Table 2. Both flooding and drought during the tillering stage reduced the maximum tiller size, indicating that flooding and drought during the tillering stage can effectively suppress the occurrence of ineffective tillers. The effective tiller rate of heavy drought (H2) treatment was the lowest, and the effective tiller rates of flooded treatments (L1, L2) and drought treatment (H1) exceeded that of CK. This indicates that heavy drought during the rice tillering period reduces the effective tiller rate, while flooding and light drought have little effect on the effective tiller rate. Flooding and maintaining a larger seepage can also increase the effective tiller rate.

**Table 2.** Characteristics of rice stem tillers for water level management at the tiller stage.

| Processing | Maximum Stem Size (Million Ears/ha) | Effective Stem Tiller Volume (Million Ears/ha) | Effective Tiller Rate (%) |
|---|---|---|---|
| L1 | 334 | 256 | 69.17 |
| L2 | 355 | 269 | 68.58 |
| H1 | 377 | 283 | 68.41 |
| H2 | 344 | 246 | 62.80 |
| CK | 388 | 276 | 63.60 |

### *3.3. Simulation of the Water Level Regulated Rice Stem Tiller Elongation Model*

The stem tiller elongation curve of rice consists of two parts: an "S"-shaped rising curve in which new tillers are dominant and an "S"-shaped falling curve in which the tiller extinction process is dominant. The basic kinetic model of stem tiller elongation in rice populations, DMOR, was derived by Wang Fuyu et al. [26], who considered the idea of using two logistic curves together. In this paper, DMOR was used to simulate stem tiller growth during the tillering stage of rice. The fitted parameters and effects of the basic kinetic model on rice stem tiller elongation during tiller water level management are shown in Table 3 and Figure 3. The statistical results show that the model has a very good fit with coefficients of determination above 0.99. When comparing the model parameter A with the corresponding measured maximum stem tiller values, it can be seen that the fitted *A* values are very close to the measured values, and the parameter $c_2$ has a large inter-annual variation, while the intra-annual variation between treatments is relatively small.

**Table 3.** Fitting results of the basic kinetic model for water level management rice stem tiller dynamics.

| Processing | Model Parameters | | | | | | | Statistical Parameters | | |
|---|---|---|---|---|---|---|---|---|---|---|
| | *A* | $b_1$ | $c_1$ | *B* | $b_2$ | $c_2$ | *C* | RMSE | R2 | F Statistics |
| L1 | 350.000 | 0.128 | 10.191 | 166.718 | 0.150 | 814.180 | 50 | 3.454 | 0.996943 | 4239.808 |
| L2 | 360.000 | 0.130 | 11.025 | 159.809 | 0.187 | 4662.877 | 50 | 3.347 | 0.997478 | 5142.022 |
| H1 | 376.952 | 0.126 | 11.068 | 161.164 | 0.185 | 6397.475 | 50 | 4.413 | 0.996157 | 3370.289 |
| H2 | 355.000 | 0.118 | 9.733 | 177.162 | 0.206 | 10,508.498 | 50 | 5.190 | 0.993485 | 1982.306 |
| CK | 390.000 | 0.145 | 15.719 | 178.929 | 0.200 | 7505.679 | 50 | 4.451 | 0.996402 | 3599.760 |

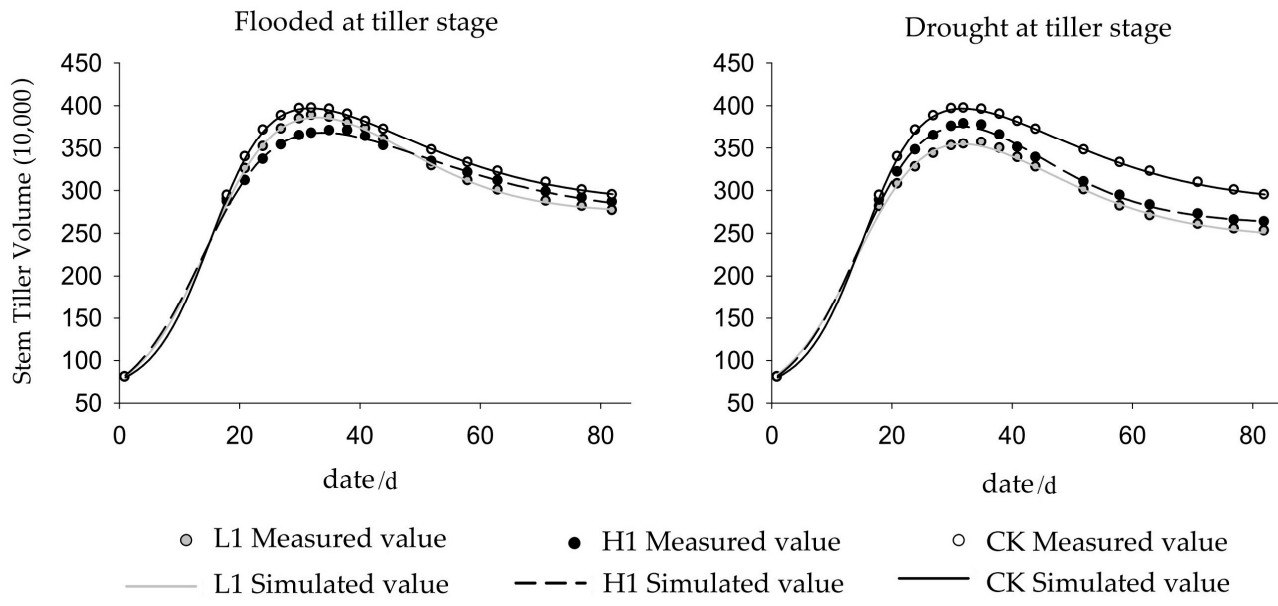

**Figure 3.** Fitted stem tiller elongation curve for rice under water level management at the tiller stage.

### *3.4. Dynamic Analysis of Water Level Regulated Rice Stem Tiller Waxing and Waning Logistic Nesting Curves*

The first- and second-order derivatives of the logistic equation can be used to analyze the growth and extinction rates of the fitted stem tiller growth and extinction curves, which can be divided into two stages: accelerated and decelerated growth and accelerated and decelerated extinction. The dynamics of the water level management rice stem tiller growth logistic curve is shown in Figure 4. It can be seen that the sequence of time when the maximum stem tiller volume levels were reached for each treatment was as follows: L1 (35 d), L2, CK (36 d), H2 (37 d), and H1 (38 d). The maximum growth rates of stem tiller growth segments in descending order were CK > L1 > H1 > L2 > H2, indicating that

flooding and drought during the tillering stage of rice reduced the growth rate of the stem tiller. The maximum extinction rates of stem tiller extinction segments were in the following order from high to low: CK > H2 > H1 > L2 > L1.

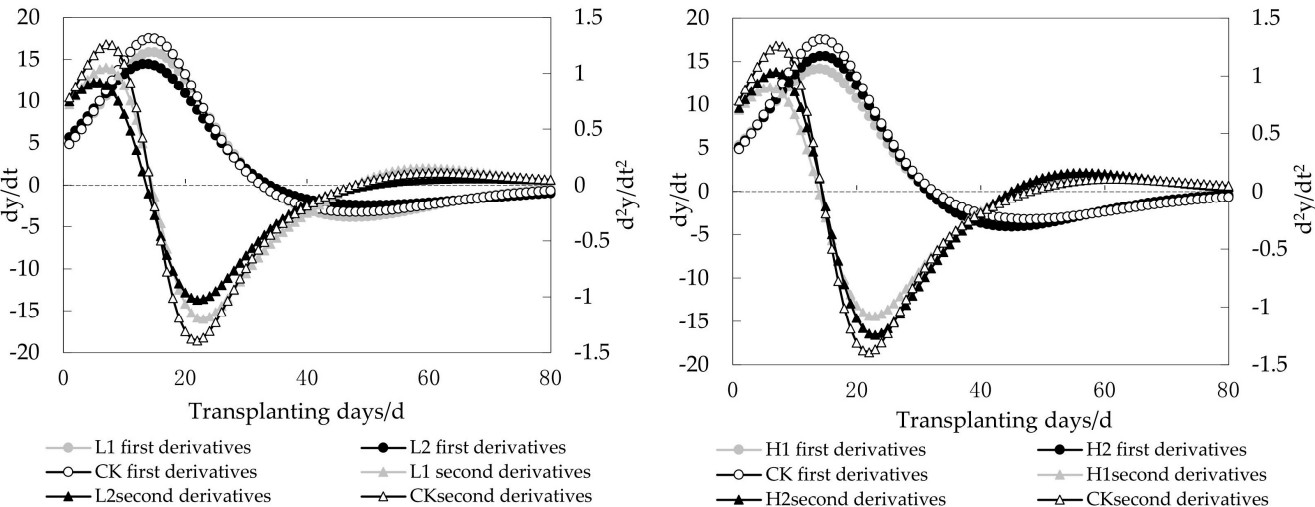

**Figure 4.** Dynamic analysis of logistic nesting curves for water level management rice stem tiller elongation.

## 4. Conclusions and Suggestions

This paper focuses on the dynamic changes, characterization, stem tiller waxing and waning model, and kinetic analysis of stem tillers subject to water level management during the rice tillering stage. The main findings are as follows.

(1) Both flooding and drought during the tillering period inhibited the level and rate of rice tillering. Compared with the control, the daily growth of rice in the flooded and drought treatments decreased more slowly and to a lesser extent, and a significant compensatory effect was observed in both drought treatments after rehydration.

(2) Both flooding and drought during the tillering stage reduced maximum tiller production, indicating that, during the tillering stage, flooding and drought can effectively suppress the occurrence of ineffective tillers, heavy drought reduced the effective tiller rate, flooding and light drought had little effect on the effective tiller rate, and flooding and maintaining a high infiltration rate increased the effective tiller rate.

(3) In this study, the basic kinetic model of tiller extinction (DMOR) was used to fit the tiller extinction process, and the statistical results show that the model had a very high fit, with coefficients of determination reaching above 0.99. At the same time, the growth and extinction rates of the stem tiller growth curve were fitted. The maximum growth rates of the stem tiller growth section were ranked as CK > L1 > H1 > L2 > H2, and the maximum extinction rates of the stem tiller extinction section were ranked as CK > H2 > H1 > L2 > L1, indicating that both flooding and drought during the tillering stage could reduce the growth and extinction rates of the tillers.

(4) This paper presents innovative water level regulation technology, coupled with water-saving irrigation and controlled drainage, which can effectively change the original irrigation and drainage mode of rice. Notably, raising the upper limit of rainwater storage can not only effectively reduce irrigation water consumption, but it also plays the role of flood control and waterlogging storage in rice fields.

(5) In future research, this topic could be approached from a microscopic perspective via studying the process of material accumulation in rice or exploring how water level management regulates the microscopic processes of nitrogen flow in rice plants.

**Author Contributions:** X.H. Writing—original draft; Y.Y. and Y.X. Writing—review & editing; F.X. Chart Editor; M.X. Model Correction. All authors have read and agreed to the published version of the manuscript.

**Funding:** This research was funded by the Fundamental Public Welfare Research Program of Zhejiang Province (LGN18E090002) and the Key R&D Program of Zhejiang (2022C02035).

**Data Availability Statement:** Not applicable.

**Conflicts of Interest:** The authors declare no conflict of interest.

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
