# Peer review of "Analysis of the Dynamics and Characteristics of Rice Stem Tillers via Water Level Management"

_water, doi:10.3390/w15061034_

Round 1

Reviewer 1 Report

After carefully read it, I found the manuscript is well expressed and in scope of the journal. However, it still needs some minor revisions before published.

1. Check the manuscript to avoid spelling and grammar errors.

2. The transitions between paragraphs are not smooth in the Introduction, please revise carefully.

3. In “experimental design” part, please supplement the basis for setting the upper and lower limits of water level regulation.

4. Please briefly state the results of the experiment, focusing on effects resulting from water level differences.

5. Add the full names of the abbreviations.

6. The conclusions need to be simplified.

7. The topic of Conclusions is presented as a summary of the work. Therefore, I encourage authors to think about what can be changed in rice culture and what are the most relevant and innovative knowledge that can be useful for the scientific community and for eventual farmers.

Reviewer 2 Report

The authors have presented an interesting paper which evaluated the effect of analysis of the dynamics and characteristics of rice stem tillers under water level management. The topic of this manuscript is very interesting. Following, I have included some comments aimed to enhance the paper:

1.      I suggest to the authors to add a new section detailing the state of the art. In this section, authors have to describe the relevant related work and their effects, authors have to identify the innovation of their study with other existing.

2.      The authors have placed some tables that are not aesthetically very visual, I suggest that they improve them.

3.      Can the authors include at the end of the introduction, more details of the objectives of their study, sine they are comparing different organic residues.

4.      Consider extending the conclusions and adding a Future works paragraph. 

5.      The bibliography is scarce and, moreover, it is not well written in many citations, et al. I beg you to improve it so that the article can be published

               Finally, we consider this work very interesting.
